# A flux tower dataset tailored for land model evaluation

Anna M. Ukkola[1], Gab Abramowitz[1], Martin G. De Kauwe[1]

[1]Climate Change Research Centre and ARC Centre of Excellence for Climate Extremes UNSW Sydney, Kensington, NSW, Australia

*Correspondence to*: Anna M. Ukkola (a.ukkola@unsw.edu.au)

**Abstract.** Eddy covariance flux towers measure the exchange of water, energy and carbon fluxes between the land and atmosphere. They have become invaluable for theory development and evaluating land models. However, flux tower data as measured (even after site post-processing) are not directly suitable for land surface

modelling due to data gaps in model forcing variables, inappropriate gap-filling, formatting and varying data quality. Here we present a quality-control and data-formatting pipeline for tower data from FLUXNET2015, La Thuile and OzFlux syntheses and the resultant 170-site globally distributed flux tower dataset specifically designed for use in land modelling. The dataset underpins the second phase of the PLUMBER land surface model benchmarking evaluation project, an international model intercomparison project encompassing >20 land surface

and biosphere models. The dataset is provided in the Assistance for Land-surface Modelling Activities (ALMA) NetCDF format and is CF-NetCDF compliant. For forcing land surface models, the dataset provides fully gap-filled meteorological data that has had periods of low data quality removed. Additional constraints required for land models, such as reference measurement heights, vegetation types and satellite-based monthly leaf area index estimates, are also included. For model evaluation, the dataset provides estimates of key water, carbon and energy

variables, with the latent and sensible heat fluxes additionally corrected for energy balance closure. The dataset provides a total of 1040 site years covering the period 1992-2018, with individual sites spanning from 1 to 21 years. The dataset is available at http://dx.doi.org/10.25914/5fdb0902607e1 (Ukkola et al., 2021).

**1 Introduction**

The global network of flux towers now encompasses >900 sites globally (https://fluxnet.org/), with the longest records spanning over three decades. With their increasing spatial and temporal coverage, flux towers have become an invaluable dataset for evaluating process representation in land surface models (LSMs). LSMs within

climate models are key tools for projecting future climates and also operate within operational weather and seasonal prediction models (Pitman, 2003; Dirmeyer et al., 2019). Their key role is to simulate the terrestrial carbon, water and energy cycles both in coupled climate models and uncoupled stand-alone applications. Flux towers provide simultaneous observations of the meteorological data needed to force offline LSMs as well as estimates of key ecosystem water, energy and carbon fluxes at a spatial scale against which LSMs can be

evaluated. Flux towers are also one of the few data sources to provide measurements at time scales appropriate for diagnosing model process representations, providing high frequency sub-daily (typically 30min) observations. As such, they have enabled model evaluation ranging from sub-diurnal to seasonal and inter-annual scales (Whitley et al., 2016; Williams et al., 2009; Wang et al., 2011; Renner et al., 2021; Blyth et al., 2010; Best et al.,



2015). Flux tower data have also been instrumental in enabling development of LSMs for extreme events such as drought (Harper et al., 2020; Ukkola et al., 2016; Martínez-de la Torre et al., 2019).

Several global multi-site collections such as FLUXNET2015 (Pastorello et al., 2020) have been released that provide valuable opportunities for evaluating LSMs across multiple climates and biomes. Whilst these collections overcome many limitations of raw flux tower data, the data are not provided in a format directly usable in land surface modelling. The datasets require varying levels of gap-filling, unit conversions and data formatting to be applicable for modelling exercises, and are missing key metadata, such as measurement height and vegetation characteristics. Most importantly, not all flux tower data releases provide temporally continuous meteorological
observations which are essential for forcing LSMs. FLUXNET2015 overcomes this key limitation by providing fully gap-filled meteorological observations but includes long periods of gap-filling at some sites, resulting in missing diurnal and/or seasonal cycles. Extended periods of synthesised meteorological variables are problematic in model applications, not only because they bias model estimates at concurrent time steps, but also because they bias future model predictions due to model state memory, such as soil moisture. As such, the data quality
requirements for land modelling present a challenge that is not yet met by standard flux tower data releases.

       Here we present a collection of 170 globally-distributed flux tower sites collated from three data releases (FLUXNET2015, La Thuile and OzFlux) that results from applying land surface model focused quality control and ancillary data collation. By combining multiple data sources, we were able to maximise the number of
available sites to enable model evaluation against a wider range of climate and vegetation conditions. The dataset covers the period 1992-2018 with individual sites spanning from 1 to 21 years, with a total of 1040 site years. The dataset provides quality-controlled, fully gap-filled meteorological variables for forcing LSMs, together with a comprehensive set of flux variables for model evaluation. The data are provided in the Assistance for Land-surface Modelling Activities (ALMA; https://www.lmd.jussieu.fr/~polcher/ALMA/) format, the international standard in
land surface modelling, and are Climate and Forecast (CF) NetCDF (https://cfconventions.org/) compliant. The dataset additionally provides various metadata for the sites, including reference / measurement height (for emulating the lowest layer of the atmospheric model to which the LSM would be coupled), vegetation type (to ensure plant physiological traits are appropriate) and two different satellite-derived estimates of each site's monthly leaf area index (LAI). The dataset underpins the second phase of the Protocol for the Analysis of Land
Surface Models (PALS) Land Surface Model Benchmarking Evaluation Project (PLUMBER; Best et al., 2015) which has participants from >20 land surface and biosphere modelling groups internationally. Whilst primarily designed for modelling purposes, the dataset would also be valuable for other applications requiring quality-controlled meteorological data at multiple sites. In the following sections we describe the processing steps to derive the dataset.


## 2 Methods

### 2.1 Datasets



We collated data for 223 flux towers from three flux tower data collections. We first obtained all available Australian sites from the OzFlux network (Isaac et al., 2017). We then obtained all Tier 1 (open data policy) globally distributed sites from FLUXNET2015 (November 2016 release; Pastorello et al., 2020), excluding sites available in OzFlux. For all FLUXNET2015 sites, data from the "FULLSET" release was used. Finally, additional sites that were not present in OzFlux or FLUXNET2015 were taken from the La Thuile Free Fair-Use release (https://fluxnet.org/data/la-thuile-dataset/). The final dataset consisted of 29 sites from OzFlux, 132 from FLUXNET2015 and 62 from La Thuile. These sites were further screened to derive the final subset of 170 sites using the protocols detailed below.

## 2.2 Processing steps

We undertook multiple processing steps to derive the final, quality-controlled dataset. The data were first pre-processed with the FluxnetLSM R package (Ukkola et al., 2017) to convert the files to ALMA-formatted NetCDF files with consistent units and variable conventions. The data were subsequently screened using expert judgement to only retain period of good quality meteorological data. Additional corrections were then made to meteorological data to remove outliers, non-physical values and gap-fill any remaining missing values. The flux variables were not screened but additional latent and sensible heat flux estimates were calculated to correct for energy balance closure. Finally, we derived two independent leaf area index time series for each site from remotely sensed data to account for uncertainties in satellite-derived LAI. A flowchart of the processing pipeline is shown in Figure 1, with each step described in detail below.

### 2.2.1 Initial processing with FluxnetLSM

The three datasets come in various formats, different units and variable naming conventions. We used the FluxnetLSM R package (Ukkola et al., 2017) which has been designed to translate flux tower data for use in land surface modelling. The package was used to process the data into ALMA-formatted CF-compliant NetCDF files with consistent variable names and units to be readily usable in land surface modelling (see Table 1 for ALMA conventions and variables included in the final dataset). In addition, FluxnetLSM was used to further gap-fill meteorological and flux variables and to include additional site metadata, such as elevation, reference and vegetation canopy heights, and vegetation type (following the International Geosphere-Biosphere Programme (IGBP) classification) in the NetCDF files. While some of the information could be obtained from Fluxnet or regional networks, we supplemented site metadata available in FluxnetLSM by extracting information from publications and site principal investigators. These metadata were collected to inform modelling choices and are included in the final NetCDF files. FluxnetLSM is fully reproducible and provides a documented framework to replace ad hoc processing methods used in many previous flux tower collections for LSMs. The version of FluxnetLSM used for processing is documented in the NetCDF file metadata.

FluxnetLSM was run separately for each parent dataset. OzFlux was first pre-processed to remove incomplete years as land surface models require whole years of data for spinning up soil water and temperature states. To achieve this, the data were first gap-filled data to complete days and incomplete years then removed using the



FluxnetLSM function "*preprocess_OzFlux*". This step was not required for FLUXNET2015 and La Thuile as they only report whole years. FluxnetLSM was subsequently used to process each dataset using the commands provided in Supplementary Section 1.

FluxnetLSM can be used to screen the data for missing and gap-filled time steps but this option was not used, instead setting the allowed level of missing and gap-filled data to 100% for all datasets and variables to allow subsequent manual visual data screening (section 2.2.2). However, the gap-filling methods for meteorological variables were set differently for each dataset. FLUXNET2015 provides continuous, downscaled ERAinterim estimates for all meteorological variables; these were used to gap-fill all missing time steps in the meteorological

variables (setting met_gap-fill to "*ERAinterim*" in FluxnetLSM). For OzFlux and La Thuile, statistical gap-filling methods provided in FluxnetLSM were used (setting met_gap-fill to "*statistical*"). For all variables except surface air pressure and incoming long wave radiation, short data gaps (up to 4 hours) were gap-filled using linear interpolation. Longer data gaps (up to 10 days for OzFlux and 365 days for La Thuile) were gap-filled using "*copyfill*" which takes the mean of the corresponding time steps during other years. Surface air pressure and

incoming longwave radiation were synthesised using empirical methods. Air pressure was calculated from air temperature and elevation using a barometric formula (Ukkola et al., 2017). Longwave radiation was calculated from air temperature and relative humidity using the method of (Abramowitz et al., 2012). The synthesised values were then used to gap-fill missing time steps.

Flux variables were gap-filled using statistical methods for all datasets. As per meteorological variables, short gaps of up to 4 hours were gap-filled using linear interpolation. Longer gaps (up to 30 days for OzFlux and FLUXNET2015, and 365 days for La Thuile) were gap-filled using a linear regression of each flux variable against incoming shortwave radiation, air temperature and humidity (relative humidity or vapour pressure deficit). In the absence of air temperature or humidity data, the linear regression was constructed against shortwave radiation

only. A separate linear model was created for day- and night-time data. Further details of all gap-filling methods can be found in (Ukkola et al., 2017).

**2.2.2 Site and time period selection**

We screened the original dataset of 223 sites to only retain sites and time periods with good quality meteorological forcing data. This was done to ensure models were forced with data that was largely observed to avoid biasing the model flux estimates. We used expert judgement to visually manually screen sites instead of an automated process to be able to compromise between data quality and time series length. During screening, we prioritised five key meteorological variables in site selection that have the largest influence on LSM simulations: incoming shortwave

radiation ($SW_{down}$), precipitation (Precip), air temperature ($T_{air}$), air humidity ($Q_{air}$) and wind speed (Wind). These variables were allowed to have approximately 10% or fewer gap-filled time steps in any given year. If no years fulfilling this criterion were available, the site was excluded. For sites with heavily gap-filled or missing periods in the middle of the time series, we chose the longest continuous period with good quality meteorological data. The remaining three meteorological variables (incoming longwave radiation ($LW_{down}$), atmospheric $CO_2$

concentration ($CO_{2\_air}$) and air pressure ($P_{surf}$)) were allowed to be gap-filled or missing for a site to be selected





but any missing or poor-quality data were later corrected as a post-processing step (section 3.2.3). Not all sites report these variables and as such, the less strict criteria were applied to retain as many sites as possible. The flux variables were not screened to allow model evaluation at multiple time scales and specific events. The specific criteria for excluding a site or time periods are provided for each site in Table S1. After site selection, the final

dataset included 23 sites from OzFlux, 102 from FLUXNET2015 and 45 from La Thuile. Table S1 provides a list of the selected sites, including the criteria for time period selection. Table S2 lists excluded sites and the reason for omitting them.

Figure 2 presents examples of how the selection criteria were applied at three sites. AU-Lit shows a site where no

adjustments to the time period were required. All key meteorological variables are largely observed, with only 3.3-5.1% of the 2-year time series gap-filled. As such, the full time series was selected for this site. BE-Bra shows an example where a subset of the years were excluded from the final dataset due to a heavily gap-filled year (2003) in the middle of the time series. During 2003, four key variables ($SW_{down}$, Precip, $T_{air}$ and Wind) are largely gap-filled, leading to unrealistic seasonal cycles in these variables. As such, the longest continuous period with low

levels of gap-filling (2004-2014) was chosen, leading to years prior to and including 2003 being discarded. US-Tw2 is an example of a site that was excluded from the final dataset. In both available years, four meteorological variables ($SW_{down}$, $T_{air}$, $Q_{air}$ and Wind) are ~50% gap-filled, exceeding our threshold of ~10%. Furthermore, no observed precipitation data were available, with the time series fully gap-filled.

**2.2.3 Further corrections to meteorological data**

After selecting the final sites, meteorological variables were further corrected for anomalous values, step changes and missing data. These corrections mainly applied to $CO_{2\_air}$ and $LW_{down}$ due a larger proportion of gap-filled and missing data in these variables. Anomalous or non-physical values in other variables were also corrected at

individual sites.

For atmospheric $CO_2$ concentration, we screened the data for step changes, unrealistically high concentrations and missing data. Where $CO_{2\_air}$ was not provided for a site, we used annual concentrations from the Mauna Loa atmospheric $CO_2$ time series (https://www.esrl.noaa.gov/gmd/ccgg/trends/mlo.html) for the period covered by

site observations. The annual values were repeated to match the site temporal resolution (half-hourly or hourly) each year. Missing $CO_2$ values were gap-filled by predicting $CO_{2\_air}$ from a linear regression of available $CO_{2\_air}$ values against time, except when large data gaps (multiple months or longer) existed in which case $CO_{2\_air}$ was replaced with the annual Mauna Loa values.

For OzFlux sites, unphysical values existed in the dataset that were corrected. These included negative Precip, $SW_{down}$, $LW_{down}$, Wind and $Q_{air}$ (vapour pressure deficit and/or relative/specific humidity) which were capped at zero. Similarly, relative humidity values above 100% were capped at 100%. At a further 11 sites, we also corrected large step changes in $CO_{2\_air}$, heavily gap-filled periods in $LW_{down}$ (which led to unrealistic seasonal cycles) and anomalous values in $P_{surf}$ and relative humidity. Table S3 summarises the corrections made to meteorological data





at each site. All corrections done during this post-processing step are documented at
https://github.com/aukkola/PLUMBER2/blob/master/functions/site_exceptions.R.

**2.2.4 Energy balance closure correction of latent and sensible heat fluxes**

Latent ($Q_{le}$) and sensible ($Q_h$) heat fluxes were corrected for energy balance closure (EBC) using the Bowen ratio
method (Mauder et al., 2020) to aid model evaluation. At flux tower sites, the sum of measured latent and sensible
heat fluxes is commonly lower than available energy (Wohlfahrt et al., 2009), complicating comparison with
models which conserve energy. The FLUXNET2015 dataset provides EBC-corrected $Q_{le}$ and $Q_h$ and as such, for
sites derived from FLUXNET2015 these estimates were used (variables LE_CORR and H_CORR in
FLUXNET2015; . For La Thuile and OzFlux sites, $Q_{le}$ and $Q_h$ were EBC-corrected using a procedure adapted
from FLUXNET2015.

The EBC-corrected fluxes were obtained by multiplying $Q_{le}$ and $Q_h$ by an EBC correction factor ($f_{EBC}$). $f_{EBC}$ was
calculated for each time step separately as $f_{EBC} = (R_{net}\text{-}G) / (Q_h + Q_{le})$ where $R_{net}$ is net radiation and G ground
heat flux (all variables are in W m$^{-2}$). Only time steps for which all four energy balance components were available
were used. The $f_{EBC}$ time series was further filtered for data quality to only retain time steps for which observed
G, and observed or good quality (qc value ≤1) $Q_h$ and $Q_{le}$ data were available. To remove outliers, $f_{EBC}$ values
outside 1.5 times the interquartile range were then discarded.

The fluxes were then corrected using a two-step method. First, for each time step, a moving window of ±15 days
was used to select $f_{EBC}$ for all time steps within the hours 22:00-2:30 and 10:00-14:30. Other times were discarded
to avoid periods of large changes in ecosystem heat storage during sunrise and sunset periods which can bias the
energy balance closure estimates (Pastorello et al., 2020). If at least five $f_{EBC}$ values were available within the
moving window, the median of these values was used to correct $Q_{le}$ and $Q_h$. Otherwise, the same moving window
of ±15 days and hours of the day was applied to the same time step using the current, previous and next year (if
available). The median of all available $f_{EBC}$ was then used to correct $Q_{le}$ and $Q_h$. If no available $f_{EBC}$ values were
found using this method, the fluxes for that time step were not corrected.

**2.2.4 Leaf area index processing**

We obtained two independent remotely sensed leaf area index (LAI) time series for each site inputs to account
for large uncertainties in satellite-derived LAI estimates (Zhu et al., 2016). The LAI time series can be used to
force LSMs that do not include a predictive carbon cycle and require prescribed LAI as an input. The standardised
LAI time series are also useful reducing the degrees of freedom in evaluation studies by allowing the models to
be driven by the same LAI estimates, and allow the minimisation of LAI-driven model errors at sites where
observed and modelled LAI converge strongly. The LAI data were derived from Moderate Resolution Imaging
Spectroradiometer (MODIS) and Copernicus Global Land Service products as these products provide long-term
records at high (≤ 1 km) spatial resolution.




### 2.2.5.1 MODIS LAI

We used the MODIS product MCD15A2H, which is derived from a combination of the Terra and Aqua sensors at 500 m spatial resolution and 8-daily temporal resolution, starting in January 2000. The LAI data and associated
standard deviation and QC flags were obtained using the R package *MODISTools* (Koen, 2020). The pixel containing the site and its surrounding pixels (in total nine pixels) were obtained for each site. Only good quality data (QC flag values 0, 2, 24, 26, 32, 34, 56 and 58) were kept and all other values were set to missing. At each time step, a weighted mean was then calculated from the nine pixels by weighting them by their standard deviation error (defined as $1/\sigma^2$). The resulting 8-daily time series were then gap-filled using a cubic spline function
(Forsythe et al., 1977) and any negative LAI values set to zero. To remove unrealistic short-term variability in LAI, e.g. due to cloud artefacts, that remained after the initial quality control, several steps were taken to further smooth the time series. The gap-filled time series was first smoothed using a cubic smoothing spline. A climatology (46 time steps) was then calculated from all available years. An anomaly time series was then created by removing the climatology and smoothed by taking a rolling mean over a window of $\pm6$ time steps to further
remove short-term variability. The climatology was then added to the smoothed time series and the 8-daily time series interpolated to the time resolution of the flux tower data, using the climatological values prior to MODIS commencing in January 2000.

### 2.2.5.2 Copernicus LAI

We used the Copernicus Global Land Service LAI v.2.0.2. which provides LAI estimates at 1 km spatial resolution and 10-daily temporal resolution for the period 1999-2017. The estimates have been derived from SPOT-VGT and PROBA-V sensors (Smets et al., 2019). The 10-daily data were first averaged to monthly by taking the maximum of the three 10-daily values for each month following the maximum composite procedure to remove
low values e.g. due to cloud contamination. The data were then smoothed spatially by averaging each pixel with its surrounding pixels (with each pixel representing the mean of nine pixels). The monthly values were then extracted for each site using the pixel containing the site. If the value for the pixel containing the site was missing, the value from the nearest non-missing pixel was used. To remove non-physical short-term variability, the monthly site time series was then smoothed using a cubic smoothing spline. A monthly climatology was then
calculated and an anomaly time series calculated by removing the climatology from the monthly LAI time series. The anomaly time series was smoothed by taking a rolling mean over a window of $\pm6$ time steps to further remove short-term variability, before adding the climatology to the smoothed anomalies. Finally, the resulting monthly time series was interpolated to the time resolution of the flux tower data, using the climatology for time periods not covered by the product.


### 2.2.5.3 LAI selection for sites

Both Copernicus and MODIS LAI were provided for each site but we selected one as a preferred LAI time series for each site to use as the default for use with LSMs that rely on prescribed LAI. Overall, we selected MODIS as
the default time series due to its higher spatial resolution but where MODIS was deemed unrealistic for the site





due to its magnitude, seasonal cycle or non-physical short-term variations, Copernicus was selected instead. Table S1 summarises the selected LAI time series for each site. The preferred LAI variable was called "LAI" in the final NetCDF files and the alternative time series "LAI_alternative".

## 3 Results

### 3.1 Global distribution of selected sites

The final dataset includes 170 globally-distributed sites shown in Figure 3a. The majority of the sites are located in North America, Europe and Australia, with 3 sites located in South America, 4 in Africa and 11 in Asia. The excluded sites are largely located in data-rich regions and as such did not significantly change the global distribution of sites. The dataset covers the periods 1992-2018, with a total of 1041 site years. Individual site records span 1 to 21 years, with a median record length of 4.5 years (Figure 3b). 39 sites cover ≥10 years and 14
sites ≥15 years.

The sites cover a wide range of biomes, ranging from grasslands and savannas to forest ecosystems (Figure 3c). The majority of sites are located in grassland (40), forested (89) and cropland (17) ecosystems. 22 sites are located in savanna and shrubland ecosystems and 10 sites in wetlands. The sites also cover a wide range of climates, with
Figure 3d showing the sites within the global range of mean annual precipitation (MAP) and mean annual temperature (MAT) from the Climatic Research Unit (CRU) TS 4.02 dataset (Harris et al., 2014). The sites capture the global climatic range well, but only a limited number of sites were available in wet tropical environments with high MAP and MAT and very cold environments (MAT < 0°C). The excluded sites lie largely within the climate envelope covered by the final dataset, thus not strongly influencing the climate range covered by the final dataset.


### 3.2 Impact of screening meteorological variables

For the selected sites, the original time series was reduced at multiple sites to exclude periods of poor-quality meteorological data. The number of years excluded at each site is shown in Figure 4. Regionally, the average
number of years excluded was similar over North America (mean: 1.8, median: 1) and Europe (1.9, 1) whereas fewer years were removed over Australia (0.7, 0) (see sub-panels in Figure 3a for region definitions). The number of excluded years was also similar across the FLUXNET2015 (mean: 2.0, median: 1) and La Thuile sites (1.3, 1) datasets but lower for OzFlux (as per Australia). Overall, there were no systematic spatial variations in the number of years excluded.


For the selected sites, our data screening reduced the mean record length by 1.7 years (median: 1), ranging from 0 to 12 years for individual sites (Figure 4b). A total of 283 site years were removed. The majority of sites (139 out of 170) had 0-2 years removed, while only 11 sites had >5 years removed. The screening also reduced the proportion of gap-filled meteorological data from 21% to 15% on average for all meteorological variables. For
the key variables, the level of gap-filled data was reduced from 10.4% to 3.6% for $T_{air}$, from 16% to 5% for Precip,



from 10.1% to 7.6% for $SW_{down}$, from 7.8% to 2.6% for $Q_{air}$ and from 15.9% to 8.3% for Wind on average across all sites. Less strict criteria were applied to $LW_{down}$ and $CO_{2\_air}$ leading to a larger proportion of gap-filled data in the final dataset. For $LW_{down}$, the proportion of gap-filled data was reduced from 48.8% to 41.5%. For $CO_{2\_air}$, the level of gap-filling remained similar (31.5% in screened data and 30.4% in the original data). This was due to the

additional gap-filling done at multiple sites to correct for step changes and a large proportion of missing data (7.2%) in the original dataset that was replaced with gap-filled values. The screening and post-processing of all meteorological variables also ensured that no missing values are present in the meteorological variables.

**3.3 Impact of energy balance closure correction on latent and sensible heat fluxes**

Flux tower observations do not commonly close the energy balance, with the sum of latent and sensible heat fluxes underestimated relative to available energy (Leuning et al., 2012; Wilson et al., 2002). This problem is particularly common in sites with heterogeneous land cover (Stoy et al., 2013) but is also driven by other factors such as unaccounted energy storage and mesoscale circulation impacts (Panin and Bernhofer, 2008; Leuning et al., 2012).

As LSMs balance all energy fluxes, latent and sensible heat fluxes were corrected for energy balance closure to aid model evaluation. In total, corrected fluxes are available for 143 sites which reported all required variables to perform the correction ($R_{net}$, G, $Q_{le}$ and $Q_h$). FLUXNET2015 already provided EBC-corrected $Q_{le}$ and $Q_h$ estimates for 82 sites and we additionally corrected 38 La Thuile sites and 23 OzFlux sites.

At the corrected sites, the instantaneous EBC (i.e. the ratio ($Q_{le}$+$Q_h$) / ($R_{net}$+G)) was 0.55 on average considering all available data points (note additional filtering was applied during correction). The EBC correction on average increased $Q_{le}$ and $Q_h$ by 25% relative to the original estimates. At individual sites, the change in $Q_{le}$ and $Q_h$ relative to uncorrected data ranged from 82% lower to 88% higher. However, for the majority of sites (123 out of 143) the correction increased $Q_{le}$ and $Q_h$.


The corrected variables should provide a more robust basis for evaluating model biases but rely on the assumption that the measured Bowen ratio is correct. Another limitation of the corrected fluxes is a larger proportion of missing data as the corrected fluxes are only provided for time steps for which the correction could be performed using our method detailed in section 3.2.4. As such, 9.2% of the corrected $Q_{le}$ is missing across all site years

compared to 1.3% in the original $Q_{le}$ estimates. Similarly for $Q_h$, 9.2% of corrected fluxes are missing compared to 0.6% in the original data.

**4 Data availability**

The final dataset is available at http://dx.doi.org/10.25914/5fdb0902607e1 (Ukkola et al., 2021). The data can also be obtained through https://modelevaluation.org/, including diagnostic plots of key variables for each site. The original flux tower datasets are available upon registration from the following websites: OzFlux (http://www.ozflux.org.au/), FLUXNET2015 (https://fluxnet.org/data/fluxnet2015-dataset/) and La Thuile (https://fluxnet.org/data/la-thuile-dataset/). MODIS LAI data can be obtained with the freely available

MODISTools R package. The remaining datasets are freely available from: Copernicus LAI



(https://land.copernicus.eu/global/products/lai), Mauna Loa $CO_2$ (https://www.esrl.noaa.gov/gmd/webdata/ccgg/trends/co2/co2_annmean_mlo.txt) and CRU TS4.02 precipitation and mean temperature (https://crudata.uea.ac.uk/cru/data/hrg/cru_ts_4.02/).

## 5 Code availability

The processing codes are available at https://github.com/aukkola/PLUMBER2.

## 6 Discussion and conclusions

We have presented a quality-controlled flux tower dataset for 170 sites for use in land surface modelling. Whilst the dataset was developed with land surface modelling in mind, it is also suitable for other applications requiring a large collection of sites with good quality meteorological data. In our site selection, we prioritised long continuous periods of high-quality meteorological observations to derive a consistent dataset across individual sites. In doing so, shorter good quality periods were discarded for some sites (e.g. Be-Bra in Figure 2); future work might revisit these choices to retain additional data periods. FluxnetLSM provides one possible reproducible tool for automated data screening to achieve this for the FLUXNET2015, La Thuile and OzFlux releases.

The meteorological data were screened and fully gap-filled using multiple criteria. This screening should allow model simulations to be produced that are less strongly biased by high levels of gap-filling and other data quality issues that affect the original data collections. We did not quality control the flux variables used for model evaluation. This was to enable model evaluation at multiple time scales, ranging from sub-daily to interannual. This also allows models to be evaluated against individual weather and climate events, such as heatwaves and drought. The lack of screening leads to a much higher proportion of gap-filled data in the flux variables which should be taken into account when selecting sites for individual applications. For example, 31% of all the $Q_{le}$ data is gap-filled, ranging from 3% to 84% at individual sites. For $Q_h$, 24% of the data is gap-filled (2-84% at individual sites). The level of gap-filling also varies strongly by variable, for example NEE estimates are on average 67% gap-filled. The level of gap-filling for individual variables can be further vary by climatic conditions, for example higher levels of observed data is often available under extreme hot than cold conditions (van der Horst et al., 2019).

Model evaluation, particularly at shorter time scales, should thus be avoided against long periods of gap-filled data. Depending on the gap-filling methods, these periods often reflect climatological conditions at the site and do not represent diurnal and seasonal variations well. This can be particularly problematic at sites with high seasonal or interannual variability in the variables of interest. Longer (daily to monthly-scale) data gaps in flux variables were gap-filled using the regression method based on $SW_{down}$, $T_{air}$ and air humidity. The quality of these gap-filled values obviously depends on how well the site fluxes can be predicted from these three variables. This method has for example been shown to predict $Q_{le}$ well under energy-limited conditions but leads to an overestimation of $Q_{le}$ under water-stressed conditions (Haughton et al., 2018a, b).



The dataset additionally provides two alternative LAI time series for each site. These can be used as inputs to those LSMs that require LAI as an input. Alternatively, they can be used to evaluate simulated LAI in those models that predict it or to verify whether model biases arise from predictive LAI feedbacks. However, it should be noted that the remotely sensed LAI estimates are highly uncertain at site scales, with large differences between
Copernicus and MODIS LAI at many sites. LAI is a key model property and has a strong influence on simulated fluxes. As such, more accurate LAI estimates would be highly valuable for constraining models. Particularly, where site-level LAI is measured, the inclusion of these data in future flux tower collections would allow to better constrain large-scale remote sensing LAI estimates used to drive models or evaluate model-simulated LAI. Additionally, the inclusion of detailed site properties in future collections would strongly benefit model
evaluation. This includes information on vegetation composition and crop cycles, disturbance events such as fire, soil properties and irrigation. Furthermore, models ideally require parameters such as reference height and canopy height to reduce model-observations mismatches arising from model inputs. Key metadata were collected from multiple sources for this data collection but the inclusion of site characteristics in future data releases would allow for more direct access to these metadata.

Finally, whilst our dataset includes a large number of globally-distributed flux tower sites, the flux tower network includes >900 sites in total. In constructing our dataset, we used the two most common global multi-site collections (FLUXNET2015 and La Thuile), supplemented by OzFlux. Whilst many flux tower sites are not freely available, regional networks such as AmeriFlux, AsiaFlux and European Fluxes Database provide additional open
policy sites that would be valuable in expanding our dataset. The current limitation with collating flux tower sites across multiple regional networks is the different data formats and standards they provide data in. Standardisation of these datasets into a common format would strongly benefit tmodelling applications, theory development and would likely lead to more widespread implementation of these data collaboratively.

**Author contribution**

All authors designed the dataset and contributed to the quality control process. A.M.U. developed the processing codes and created the final dataset. A.M.U. prepared the manuscript with contributions from all co-authors.

**Competing interests**

The authors declare that they have no conflict of interest.

**Acknowledgements**

A.M. Ukkola, M.G. De Kauwe and G. Abramowitz acknowledge support from the Australian Research Council (ARC) Centre of Excellence for Climate Extremes (CE170100023). M.G. De Kauwe acknowledges support from the ARC Discovery Grant (DP190101823) and the NSW Research Attraction and Acceleration Program (RAAP). A.M. Ukkola is supported by the ARC Discovery Early Career Researcher Award (DE200100086). We thank the
National Computational Infrastructure at the Australian National University, an initiative of the Australian



Government, for hosting the final dataset. This work used eddy covariance data acquired and shared by the FLUXNET community, including these networks: AmeriFlux, AfriFlux, AsiaFlux, CarboAfrica, CarboEuropeIP, CarboItaly, CarboMont, ChinaFlux, Fluxnet-Canada, GreenGrass, ICOS, KoFlux, LBA, NECC, OzFlux-TERN, TCOS-Siberia and USCCC. The ERA- Interim reanalysis data are provided by ECMWF and processed by LSCE.

The FLUXNET eddy covariance data processing and harmonisation was carried out by the European Fluxes Database Cluster, AmeriFlux Management Project and Fluxdata project of FLUXNET, with the support of CDIAC and ICOS Ecosystem Thematic Center, and the OzFlux, ChinaFlux and AsiaFlux offices.

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



**Table 1: Variables provided in the final dataset (NB. not all sites provide all flux variables). Variable naming conventions follow the ALMA format where available.**

| Variable | Description | CF standard name | Unit |
|---|---|---|---|
| ***Meteorological:*** | | | |
| Precip | Precipitation rate | precipitation_flux | mm s$^{-1}$ |
| Tair | Near surface air temperature | air_temperature | K |
| SWdown | Downward shortwave radiation | surface_downwelling_shortwave_flux_in_air | W m$^{-2}$ |
| LWdown | Downward longwave radiation | surface_downwelling_longwave_flux_in_air | W m$^{-2}$ |
| Qair | Near surface specific humidity | specific_humidity | kg kg$^{-1}$ |
| VPD | Vapour pressure deficit | water_vapor_saturation_deficit_in_air | hPa |
| RH | Near surface relative humidity | relative_humidity | % |
| Wind | Scalar windspeed | wind_speed | m s$^{-1}$ |
| Psurf | Surface air pressure | surface_air_pressure | Pa |
| CO2air | Near surface $CO_2$ concentration | mole_fraction_of_carbon_dioxide_in_air | ppm |
| ***Flux:*** | | | |
| Rnet | Net radiation | surface_net_downward_radiative_flux | W m$^{-2}$ |
| SWup | Upward shortwave radiation | surface_upwelling_shortwave_flux_in_air | W m$^{-2}$ |
| Qle | Latent heat flux | surface_upward_latent_heat_flux | W m$^{-2}$ |
| Qh | Sensible heat flux | surface_upward_sensible_heat_flux | W m$^{-2}$ |
| Qg | Ground heat flux | surface_downward_heat_flux | W m$^{-2}$ |
| Qle_cor | Energy balance corrected latent heat flux | surface_upward_latent_heat_flux | W m$^{-2}$ |
| Qh_cor | Energy balance corrected sensible heat flux | surface_upward_sensible_heat_flux | W m$^{-2}$ |
| Ustar | Friction velocity | - | m s$^{-1}$ |
| NEE | Net ecosystem exchange of $CO_2$ | surface_net_downward_mass_flux_of_carbon_dioxide_expressed_as_carbon_due_to_all_land_processes_excluding_anthropogenic_land_use_change | µmol m$^{-2}$ s$^{-1}$ |
| GPP | Gross primary productivity of $CO_2$ | gross_primary_productivity_of_carbon | µmol m$^{-2}$ s$^{-1}$ |
| GPP_se* | Standard error of GPP | - | µmol m$^{-2}$ s$^{-1}$ |
| GPP_DT* | Gross primary productivity of $CO_2$ from day-time partitioning method | gross_primary_productivity_of_carbon | µmol m$^{-2}$ s$^{-1}$ |
| GPP_DT_se* | Standard error of GPP_DT | - | µmol m$^{-2}$ s$^{-1}$ |



| Resp | Ecosystem respiration | - | $\mu mol\ m^{-2}\ s^{-1}$ |
|---|---|---|---|
| Resp_se* | Standard error of Resp | - | $\mu mol\ m^{-2}\ s^{-1}$ |

*FLUXNET2015 only



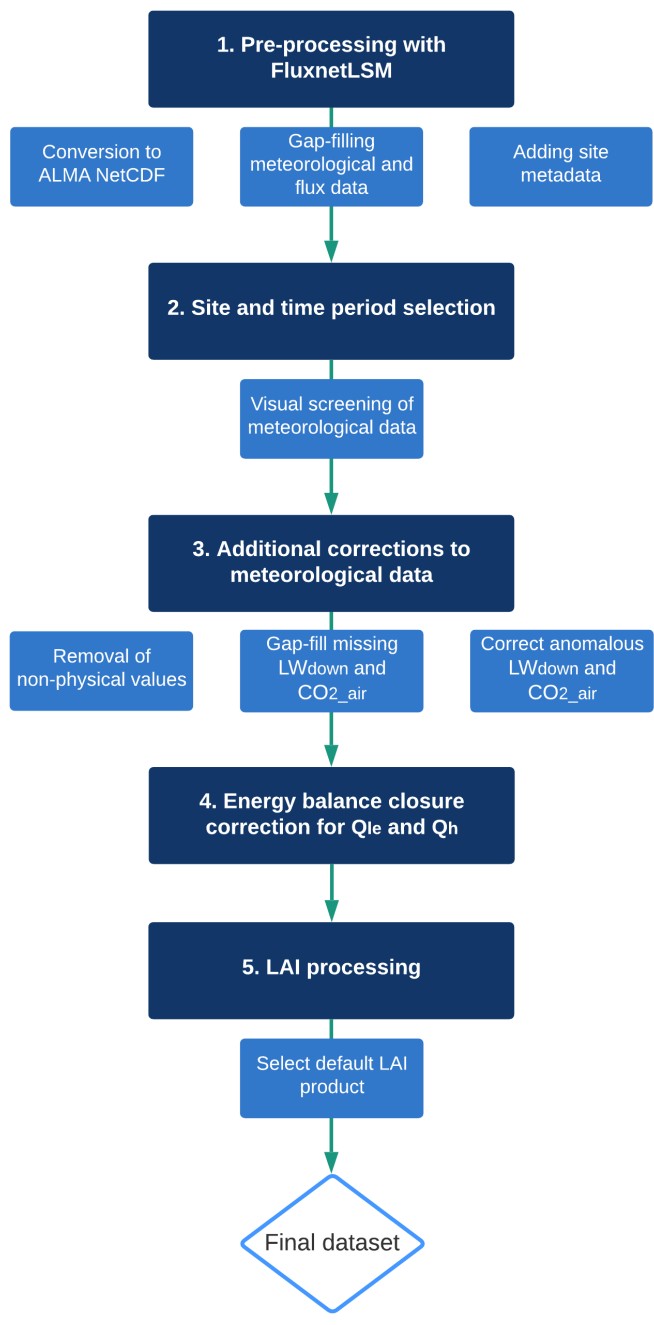

**Figure 1: A flowchart describing the data processing pipeline. The dark boxes show the main data processing steps, with the lighter boxes detailing the actions taken within each main step.**



**Figure 2: Examples of meteorological data screening for three sites (AU-Lit, BE-Bra and**
5    **US-Tw2)**

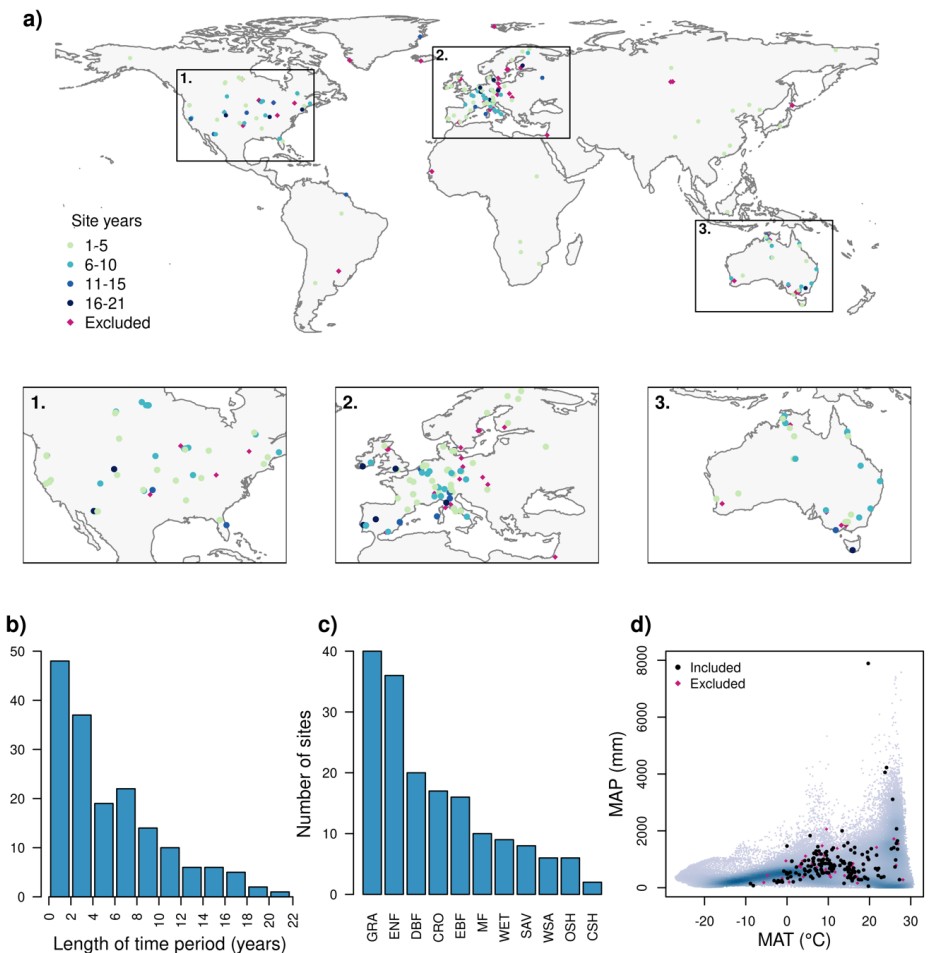

**Figure 3: Selected and excluded sites. a) a map of selected sites including the length of data period, and excluded sites, b) a histogram of record length for selected sites, c) number of selected sites per IGBP vegetation class and d) the distribution of selected and excluded sites within the global envelope of mean annual temperature (MAT) and mean annual precipitation (MAP).**



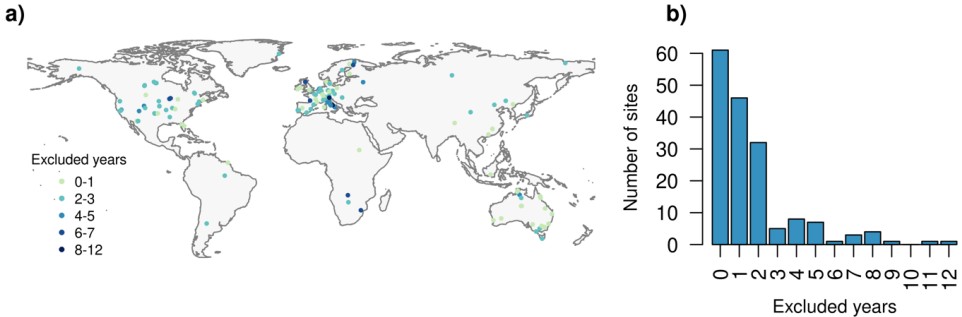

**Figure 4: Excluded years from selected sites. a) a map of selected sites showing the number of excluded years. b) a histogram of excluded site years.**

