# Peer review of "A flux tower dataset tailored for land model evaluation"

_Earth System Science Data, 2021_

## Author Response (AR1)

*RC1: This manuscript describes a set of 170 Tower Sites for use in land modeling. The set of sites have been quality-controlled and gap-filled, providing a consistent set of data that will be of wide utility to land modelers and other researchers. I commend the authors for their valuable work.*

*The paper is well-written, clear, and concise and as such I do not have much by way of critical comments. I find the manuscript to be acceptable for publication after minor revisions. This is/will be a valuable resource for the community.*

We appreciate these positive comments on the usefulness of this work. We deal with each issue raised below in turn.

*RC1: Are there any plans to update the datasets going forward with new data as it is collected or with additional sites? The authors mention that other sites could potentially be included from the greater global Tower Site network, which partially answers this question, but it doesn't resolve the question about whether / how these sites will be updated as more data is collected.*

Yes, this is of course critically important. Indeed since this manuscript has been submitted we have begun discussions with Dario Papale, Martin Jung (Fluxnet / CoCO2) and Trevor Keenan (Ameriflux) to incorporate the data processing we've outlined here into a new branch of Fluxnet's automated site data processing chain, sending appropriately formatted site data directly to modelevaluation.org. We have amended the text in the discussion/conclusion section to make this clear:

*"Finally, whilst our dataset includes a large number of globally-distributed flux tower sites, the flux tower network includes >900 sites in total. In constructing our dataset, we used the two most common global multi-site collections (FLUXNET2015 and La Thuile), supplemented by OzFlux. Whilst many flux tower sites are not freely available, regional networks such as AmeriFlux, AsiaFlux and European Fluxes Database provide additional open policy sites that would be valuable in expanding our dataset. The current limitation with collating flux tower sites across multiple regional networks is the different data formats and standards they provide data in. To this end, active discussions are underway with Fluxnet and Ameriflux organisers to incorporate the data processing and formatting detailed in this paper into their automated data processing streams, reducing duplication and lag time for the ecological and modelling community. Standardisation of these datasets into a common format would strongly benefit the wider community applications, theory development and would likely lead to a greater uptake of these data."*

*RC1: On the Flux datasets, there are variables with a _uc designation. The long name says that this is "Qh_cor joint uncertainty". I couldn't find anywhere in the text that explains what this variable is. A brief explanation in the text about what these variables are and how they were calculated would be useful.*
*Also on the Flux datasets, a few variables have _se or _DT suffixes. The se refers to standard error, but I'm not sure exactly what error is being referred to .. measurement error or error induced by gap-filling or ???*
*Several of these variables also have long names that say something like "GPP of CO2 standard error (variable ustar, nighttime partitioning)". I'm not sure what is meant by variable ustar in this context. I can guess at nighttime partitioning, but it would be a guess. As above, some additional text in the main document explaining these variables would be helpful.*

These are standard output FLUXNET2015 variables that have been included verbatim (apart from temporal processing). Some were originally not included in the variable table, but on reflection their omission was a mistake. They have now been included. We discussed explicitly providing their derivation, but given that we do not actually calculate these variables for this work, and more detail would require reference to a considerable suite of other Fluxnet variables that are otherwise not relevant to what we're presenting, we took a different

approach to resolving this issue. Table 1 now includes an additional column with the Fluxnet name for each included variable, and the caption of Table 1 provides a reference to the formal Fluxnet documentation for the calculation of these variables. As you will find in the FLUXNET2015 documentation, "nighttime partitioning" (see Reichstein et al., 2005) refers to using night time measurements to estimate ecosystem respiration (ER), and by extrapolating to daylight hours (as a function of temperature), GPP can be inferred as the difference between the total NEE flux and ER; "variable ustar" refers to uncertainty estimation by way of multiple approaches to calculating ustar, and using the resulting NEE spread as an uncertainty estimate.

> *RC1: Regarding the corrections for energy balance closure, I would like to direct your attention to this paper.*
> *Swenson, S.C., D.M. Lawrence, and S. Burns, 2019: The Impact of Biomass Heat Storage on the Canopy Energy Balance and Atmospheric Stability in the Community Land Model. JAMES, doi.org/10.1029/2018MS001476.*
> *In that paper, we found that when we included biomass heat storage in CLM, the need to correct the observed latent and sensible heat fluxes (at least during middle of day) seemed to go away (i.e., the model produced an LH+SH/Rnet ratio that was consistent with the uncorrected LH and SH), especially for forested sites. See in particular Figures 10 and 11 which show good agreement of model and obs with biomass heat storage against the uncorrected LH and SH.*
> *Note that we weren't seeing such a big discrepancy between Rnet and LH+SH, at least compared to the numbers you state in your paper (average of 0.55 for LH+SH/Rnet+G). We were seeing an average of more like 0.75 (without the G term, which I think is generally pretty small). Anyway, you do mention unaccounted energy storage as one source of the lower LH+SH vs Rnet, but then it seems like you ignore the possibility that energy storage might be playing a role with the correction that you apply. The results from our paper suggest that biomass heat storage potentially shouldn't be ignored when making the correction, especially for forest sites. Anyway, this is a rambling comment and this really isn't my area of expertise, but I wanted to raise your attention to it. What to do is unclear to me, but maybe at least some additional discussion is warranted about the assumptions and implications and from the corrections. If the results in our paper are correct, and are being interpreted correctly, then including the correction could actually degrade the accuracy of the observed fluxes rather than improve them, leading modelers to target the wrong flux values.*

Yes, this might well be the case, Ray Leuning did something similar at Tumbarumba some years ago that showed similar results. Our recollection is that applying a parametrisation that works well universally across all sites was not possible, or at least not yet.

To be clear though, that's not what we were trying to do here. We were not trying to do anything more than replicate the original Fluxnet2015 energy balance correction process for sites that were not part of that release. The issue you mention may ultimately be part of a better approach to doing this, but given that a correction was applied by Fluxnet to most of the sites as part of the 2015 release, using a relatively sophisticated approach derived by those who work most closely with these data, we thought it best to simply apply their approach to those sites that had no existing correction. The dataset allows the user to use the raw data and assume a biomass heat storage as you tested with CLM, if they wish.

> *RC1: Line 30, page 4: "We used expert judgement to visually manually screen sites instead of an"; visually or manually or both?*

Maybe tautologous, yes... we removed the word "visually".

> *RC1: Do ANY of the sites report LAI (I thought some did, but not sure about that) and if they do, did you check them against the LAI from MODIS?*

Yes, some sites do for some instants in time. For example, where LAI values are available via the BADM file, these are single values and it is not always clear if these represent a site mean or maximum value, or over what timeframe. Either way, these values are insufficient to drive a LSM. Undoubtedly LAI is likely one of the biggest uncertainties here, but given that this dataset was derived for the PLUMBER2 experiment we wanted to have a uniform approach. Site data that we did have informed the choice between the two remotely sensed products. We have modified the section on LAI selection to make this more explicit:

*"Overall, we selected MODIS as the default time series due to its higher spatial resolution but where MODIS was deemed unrealistic for the site due to its magnitude, seasonal cycle or non-physical short-term variations, using site data where available, Copernicus was selected instead."*

*RC2: This manuscript synthesizes a global network of site level meteorological and flux variables that draws from the existing flux tower networks of FLUXNET2015, OzFlux and LaThuile. For meteorological data the FLUXNET2015 data product is gap-filled using ERAinterim downscaled estimates, whereas OzFlux and LaThuile provided statistical gap filling techniques. For flux variables all data sets used statistical gap-filling approaches. If gap filling exceeds 10% of the data set, then the entire year is thrown out. A single format gap-filled data site level data product is helpful for a regional or global network of land model simulations, as well as for model-data fusion (data assimilation) purposes where a network of data could be ingested into a model to provide a regional analysis. Synthesized data sets like these are certainly welcome to the modeling and data assimilation community. This reviewer would like to encourage the authors to think about more quantitative inclusion of uncertainty with the gap-filling routines. Robust uncertainty estimates are an often overlooked but important piece included within data products both for model validation and model-data fusion exercises. See next section for more detailed feedback.*

We are obviously pleased that the reviewer thinks this work will be useful for the community.

*RC2: Sections 2.2.1: Would prefer a little more detail on the flux variable gap filling approach, where statistics on the skill of the filling methodology (linear regression against shortwave, temp and humidity) could be included from Ukkola et al., 2017?*

A detailed analysis of the performance of this particular approach is a little out of scope for this paper, given this approach did not play a major role here. Nevertheless we appreciate the need to allow readers to understand exactly what has been done. There are several existing papers that do explicitly focus on the skill of this approach, so we have rewritten this section to briefly summarise their results and include references to these studies:

*"Flux variables were gap-filled using statistical methods for all datasets. As per meteorological variables, short gaps of up to 4 hours were gap-filled using linear interpolation. Longer gaps (up to 30 days for OzFlux and FLUXNET2015, and 365 days for La Thuile) were gap-filled using a linear regression of each flux variable against incoming shortwave radiation, air temperature and humidity (relative humidity or vapour pressure deficit). This approach was demonstrated to outperform a range of LSMs in a broad range of metrics in out of sample tests (see Abramowitz, 2012; Best et al, 2015). In the absence of air temperature or humidity data, the linear regression was constructed against shortwave radiation only. A separate linear model was created for day- and night-time data. Further details of all gap-filling methods can be found in (Ukkola et al., 2017)."*

*RC2: Table 1: Isn't NEE uncertainty reported in FLUXNET2015, in addition to GPP and ER partitioning uncertainty? Why not provide here?*

Yes indeed it is, and in fact it was provided here, but omitted from the variable table in the manuscript. This has now been rectified, in addition to providing the original Fluxnet names for each of these variables and a reference to Fluxnet documentation, so that readers can find their definition and derivation in the Fluxnet database.

*RC2: Figure 2: It appears for the site BE-Bra that SW_down and T_air are labeled as gap-filled (red-line), because in the text it is mentioned that year 2003 is removed for that site. This should also be stated in the figure caption and this time period should not be listed as gap-filled (red line). US-Tw2 is same thing, state in caption this was discarded.*

Yes, this could be confusing. We have amended the caption to clearly state that these plots are pre-screening, and that some section were removed:

*"Figure 2: Examples of meteorological data pre-screening plots for three sites (AU-Lit, BE-Bra and US-Tw2). For each site different processing approaches were used and sections of this data discarded."*

> *RC2: "The majority of sites are located in grassland (40), forested (89) and cropland (17) ecosystems. 22 sites are located in savanna and shrubland ecosystems and 10 sites in wetlands."     It would be preferable to see a more specific breakdown of these sites into plant species, or perhaps into more specific classifications (e.g C3 or C4 grasses, boreal/temperate evergreen, deciduous, crop types etc). Figure 3c has some of this information, but it would be helpful to spell it out here a bit more in the text, or at least refer to Figure 3c at this point.*

From an ecological study perspective this is of course true, but keep in mind that this data set is being produced primarily for a study evaluating land models that are run inside climate models, that typically select vegetation characteristics based on a global grid of vegetation types. Many of these models would be unable to utilise this information, in particular the DGVMs. Nevertheless, where this information was easily accessible we already did include it, and indeed some soil and disturbance information, using the "vegetation_description", "soil_type" and "site_description" global attributes in the final netcdf files. Rest assured if this information were universally available for all of these sites we would have included it. Information on the C4 fraction at fluxnet sites is a long standing knowledge gap, with previous multi-site syntheses replying on satellite derived products to attempt to bridge this uncertainty (e.g. Medlyn et al. 2017 New Phyt.; De Kauwe et al. 2017 Biogesci.).

> *RC2: Figure 3:    It is helpful to show the coverage of the sites in terms of MAT and MAP, and spatially. Might it also be useful to present site level location in terms of regions with the most productivity/biomass?  Presenting a climate envelope is not as necessarily important as perhaps locating regions which have the most influence upon carbon cycle, for example.  T.*

That really depends on the purpose of a particular study. As noted above, this data set is being produced primarily for a study evaluating land models that are run inside climate models that are primarily focused on climate biophysics. Consequently, Figure 3 simply represents an overview of where sites are drawn from and as there is no attempt to carry out detailed analysis here, information about maximum productivity (etc) would be beyond the scope or aims of this paper.

> *RC2: "Model evaluation, particularly at shorter time scales, should thus be avoided against long periods of gap-filled data."  This statement could be made more quantitative by providing uncertainty bounds with the linear regression model so that the user, can make a quantitative evaluation of the model run.  Simply, providing a recommendation that long gap-filled time periods should be avoided seems a bit qualitative and not as helpful as providing an uncertainty estimate.*

It's a little unclear how one might make sure of such a quantification, except perhaps in a data assimilation exercise (which is not the purpose of the study for which this data set was derived). Either way, given that only a fraction of the gap-filling in this data set was performed by us, we are unable to derive this. While the statement could potentially be made more quantitative, it still serves an important qualitative role. We have opted not to amend the text.

> *RC2: " it should be noted that the remotely sensed LAI estimates are highly uncertain at the site scales, with large differences between Copernicus and MODIS LAI at many sites."  Could you discuss/quantify what is meant as 'highly uncertain' at site scales, and where does this uncertainty derive from?  Is it a case of algorithm uncertainty directly from the MODIS or Copernicus raw data – or is it a case of representation mismatch between the coarse product spatial resolution and the footprint of the site (1 km2)?*

It is indeed both of these. We've added some additional text to make this clearer:

*"However, it should be noted that the remotely sensed LAI estimates are uncertain at site scales, with large differences between Copernicus and MODIS LAI at many sites. This is both because of the difficulties inherent in estimating LAI from satellites (methodological) and the fact the satellite data may be drawn from a different footprint from the one that influences the  site scale measured fluxes (De Kauwe 2011 RSE)."*

> *RC2: Table S1:  The manuscript states the flux data covers the period 1992-2018, but the vast majority of sites are (FLUXNET2015) and thus only available through 2014.   I guess that is to be expected given you are drawing from FLUXNET2015, but a bit misleading.*

We have made this clearer where stated in the introduction:

*"The dataset covers the period 1992-2018 (although the majority of site records end in 2014) with individual sites spanning from 1 to 21 years, with a total of 1040 site years."*